# Participation of the miR-22-HDAC4-DLCO Axis in Patients with COPD by Tobacco and Biomass

**DOI:** 10.3390/biom9120837

**Published:** 2019-12-06

**Authors:** Yadira Velasco-Torres, Víctor Ruiz, Martha Montaño, Rogelio Pérez-Padilla, Ramcés Falfán-Valencia, Julia Pérez-Ramos, Oliver Pérez-Bautista, Carlos Ramos

**Affiliations:** 1Biological and Health Sciences, Universidad Autónoma Metropolitana-Xochimilco (UAM-X), Mexico City 04960, Mexico; yajo19@yahoo.com.mx; 2Molecular Biology Laboratory, Department of Research in Pulmonary Fibrosis, Instituto Nacional de Enfermedades Respiratorias Ismael Cosío Villegas (INER), Mexico City 14080, Mexico; vicoruz@yahoo.com.mx; 3Cellular Biology Laboratory, Department of Research in Pulmonary Fibrosis, Instituto Nacional de Enfermedades Respiratorias Ismael Cosío Villegas (INER), Mexico City 14080, Mexico; mamora572002@yahoo.com.mx; 4Department of Research in Smoking and COPD, Instituto Nacional de Enfermedades Respiratorias Ismael Cosío Villegas (INER), Mexico City 14080, Mexico; perezpad@gmail.com; 5HLA Laboratory, Instituto Nacional de Enfermedades Respiratorias Ismael Cosío Villegas (INER), Mexico City 14080, Mexico; rfalfanv@iner.gob.mx; 6Division of Biological and Health Sciences, Universidad Autónoma Metropolitana-Xochimilco (UAM-X), Mexico City 04960, Mexico; jperez@correo.xoc.uam.mx

**Keywords:** biomass smoke, copd, hdac4, dlco, mir-22, tobacco smoking

## Abstract

Chronic obstructive pulmonary disease (COPD) is characterized by airflow limitation and systemic inflammation. The main causes of COPD include interaction between genetic and environmental factors associated with tobacco smoking (COPD-TS) and/or exposure to biomass smoke (COPD-BS). Several microRNAs (miRNAs) control posttranscriptional regulation of COPD-TS associated gene expression. The miR-22-HDAC4-IL-17 axis was recently characterized. It is still unknown, however, whether this axis, participates in COPD-BS. To investigate, 50 patients diagnosed with severe-to-very severe COPD GOLD (Global Initiative for Chronic Obstructive Lung Disease) stages III/IV, were recruited, 25 women had COPD-BS (never smokers, exposed heavily to BS) and 25 had COPD-TS. Serum levels of miRNA-22-3p were measured by RT (Reverse Transcription)-qPCR, while the concentration of HDAC4 (Histone deacetylase 4) was detected by ELISA. Additionally, we looked for association between serum HDAC4 and DLCOsb (Single-breath diffusing capacity of the lung for carbon monoxide), as % of predicted by age, height, and gender, one of the main differences described between COPD-BS and COPD-TS. Women with COPD-BS were older and shorter and had a higher DLCOsb %P (percent predicted) compared to COPD-TS. Serum miR-22-3p was downregulated in COPD-BS relative to COPD-TS. In contrast, the concentration of HDAC4 was higher in COPD-BS compared to COPD-TS. Furthermore, a positive correlation between serum HDAC4 levels and DLCOsb %P was observed. We concluded that the miR-22-HDAC4-DLCO axis behaves differently in patients with COPD-BS and COPD-TS.

## 1. Introduction

The main risk factor for the development of chronic obstructive pulmonary disease (COPD) is exposure to tobacco smoke (COPD-TS). Tobacco smoke affects men and women, although available evidence suggests that women are more susceptible to the effects of TS than men [1]. However, in developing countries such as Mexico, exposure to biomass smoke is the main means of exposure for women, especially that produced by the combustion of wood [2], and produces a significant number of cases of COPD due to biomass smoke exposure (COPD-BS). 

The pathogenesis of COPD begins when the particles of tobacco or biomass smoke are inhaled and deposited deep in the airways and lungs, triggering an inflammatory response, which is perpetuated, evolving towards chronic and irreversible damage that could lead to bronchiolitis and emphysema [3]. Nevertheless, in high-resolution CT scans [4], COPD-BS patients display severe thickening of the bronchial walls, bronchiectasis, and mild or no emphysema. In contrast, COPD-TS patients have a significantly higher emphysema index with milder changes in airways [5,6]. Autopsy studies in smokers with COPD showed more emphysema and goblet cell metaplasia than women exposed to BS [7]. On the other hand, women exposed to BS presented more local scarring and pigment deposition in the lung parenchyma, and more fibrosis. These differences have suggested the existence of two clearly distinct phenotypes of COPD, one linked to tobacco smoke and another to biomass [7].

Emphysema is caused by multiple interactions between cellular and humoral immunity, a dysregulated autophagy, which is a critical point for the formation of emphysema due to the decrease in histone deacetylase activity (HDAC) [8]. These proteins are regulated by miRNAs [9,10,11], small non-coding RNAs that silence and regulate the post-transcriptional modification of mRNAs in physiological and pathological processes [12]. Recently, in vitro studies have proposed the participation of the miR-22-HDAC4-IL-17 axis in the pathogenesis of emphysema in rats exposed to tobacco smoke through miR-22 activated lung antigen presenting cells (APC) after exposure to cigarette smoke. Upregulation of miR-22 was presented in the lung APCs of smokers with emphysema, contrary to control subjects without emphysema. In rats exposed to cigarette smoke that developed emphysema, an accumulation of TH17 cells, macrophages, and neutrophils in their lungs was observed. A main target of miR-22 in APCs is the gene that encodes histone deacetylase HDAC4 [13].

Based on these findings, the objective of our study was to compare the participation of the miR-22-HDAC4-DLCO axis in individuals with COPD associated with biomass smoke inhalation (COPD-BS) to smokers with COPD (COPD-TS). Specifically, we aimed to estimate the mean difference in the expression of miR-22 and HDAC4.

## 2. Material and Methods

### 2.1. Study Population

This cross-sectional study was approved by the Science and Bioethics Committee at the Instituto Nacional de Enfermedades Respiratorias Ismael Cosío Villegas (INER) in Mexico City, a referral center of respiratory diseases in Mexico City (Protocol INER: B15 15, approved: 1 may 2015) mainly dedicated to the care of uninsured individuals. Participants were recruited at the COPD clinic of the INER, from a cohort that was followed regularly. We included individuals with COPD in GOLD (Global Initiative for Chronic Obstructive Lung Disease) stages III-IV; where in general, the prevalence of emphysema is higher than in earlier stages. We selected 25 women with COPD-BS and 25 with COPD-TS with a diagnosis confirmed spirometrically, and classified according to GOLD FEV_1_/FVC ratio (Forced Expiratory Volume in first second/Forced Vital Capacity ratio) post bronchodilator <0.7), considering the symptoms and the deterioration of the health status of the patient. Patients with COPD-TS were active smokers or had quit smoking for less than 1 year prior to be enrolled in the study, and were not exposed to biomass smoke. We excluded patients with symptoms or clinical signs of a COPD exacerbation 6 months prior to study enrollment or lower respiratory tract infection 4 weeks prior to enrollment, but also patients diagnosed with asthma, bronchiectasis, or lung cancer. Demographic, anthropometric, and clinical data of all women were collected through an interview and questionnaires. None of the patients with COPD-BS was a smoker or had been exposed to TS, all biomass patients had used wood as fuel. 

### 2.2. Pulmonary Function Testing

The diagnosis of COPD in all women was established, COPD was classified according to the exposure history following the procedures recommended by the American Thoracic Society/European Respiratory Society [1], and the standard reference for Mexicans [14]. All had post-bronchodilator airflow obstruction, a forced expiratory volume in the 1st second (FEV_1_) divided by a forced vital capacity (FVC), FEV_1_/FVC <70% ratio. FEV1 and FVC were expressed as percentage of predicted (FEV_1_%P, FVC%P). Tests were done using a Sensormedics dry seal spirometer (Yorba, Linda, CA, USA) following international ATS/ERS (American Thoracic Society/European Respiratory Society) standard [15].

DLCOsb %P (Single-breath diffusing capacity of the lung for carbon monoxide % predicted) were performed using the commercial equipment Easy One Pro and Easy One Pro Lab (Ndd^®^, Zurich, Switzerland). EasyOne Pro^®^ measures flow, tracer gas, and CO with individual sensors that fulfill the required specifications according to the ATS/ERS standard [15]. We reviewed all DLCOsb maneuvers to analyze the quality of each one in accordance with the new 2017 ERS/ATS standards. 

### 2.3. Blood Samples

Blood samples were collected in anticoagulant-free tubes (5 mL) (BD vacutainer, Becton, Franklin Lakes, NJ, USA), following the standard procedures at INER. The samples were centrifuged at 5000× *g* at room temperature for 15 min to obtain the serum and worked on the same day. For continuous storage, the serum was divided into aliquots and frozen at −80 °C until use.

### 2.4. Isolation of Serum microRNA 

The extraction of the miRNAs was performed using the miRNeasy serum/plasma kit (Qiagen, Hilden, Germany), following the manufacturer’s instructions. Aliquots of 200 μL of serum were transferred to 2 mL tubes. Next, the QIAzol Lysis Reagent (Qiagen, Valencia, CA, USA) was added, followed by 3.5 μL of spike as a control (1.6 × 10^8^ copies/μL) and 200 μL of chloroform, and then were centrifuged for 15 min at 12,000 g at 4 °C. The aqueous phase was separated, 1.5 volumes of 100% ethanol were added, and then an aliquot of 700 μL was passed through an RNeasy MinElute Spin Column centrifuge column (Qiagen) and centrifuged at 8000× *g* for 15 min. Subsequently, 700 mL of buffer RWT (RWT wash buffer; Qiagen, Valencia, CA, USA) was added to the column, followed by centrifugation for 15 s at 8000× *g* and 500 μL of buffer RPE (RPE buffer for washing membrane-bound RNA; Qiagen, Valencia, CA, USA) were added. The resulting miRNA was eluted with 20 μL of RNase-free water by centrifugation at 10,000× *g* for 1 min. The miRNA was quantified using the Nanodrop system, and the integrity was evaluated using the Agilent Bioanalyzer 2100 system (Agilent Technologies, Santa Clara, CA, USA).

### 2.5. Differential Expression of miRNA-22 in Serum by PCR Arrays

The differential expression analysis was performed for 96 miRNAs using MiScript miRNA PCR Array (MIHS-106Z; Qiagen, Valencia, CA, USA), isolated from serum. After obtaining all raw results, the data was analyzed in the Qiagen software (Data analysis file for miScript miRNA PCR Array All miRNA QC; Qiagen, Valencia, CA, USA) (available at https://www.qiagen.com/us/shop/genes-and-pathways/data-analysis-center-overview-page/). The software uses the 2-ΔCt (2^- CT gene of interest − CT internal control^) method, which analyzed and compared two specific groups using Student’s *t*-test. The data is expressed as fold change = relative quantification of 2-ΔCt for miR-22. 

### 2.6. Validation of Samples by TaqMan RT-qPCR

The cDNA was obtained from serum using the RT (Reverse Transciption) kit and TaqMan Universal Master Mix II with UNG (Uracil-N-glycosylate; Applied Biosystems-Thermo Fisher Scientific). Pre-designed commercial assay of TaqMan probes from Thermo Fisher Scientific were specific to miR-22 (hsa-miR-22-3p ID 000398). The expression level of miRNA was evaluated using the comparative threshold cycle method (ΔCt) and was normalized with a corresponding sequence of *C. elegans* miRNA as an exogenous normalizer in gene expression (spike-in cel-miR-39). The relative concentration of miR-22 was described as a fold change (2-ΔCt), by using the equation ΔCt = (Ct miRNA-Ct spike). The cut-off value was established as cycle ≤40, and it was considered that a gene was not detectable when Ct >40 and the signal was below the established limits [16,17].

### 2.7. Protein Quantification

The search for the HDAC4 target of miR-22 in lung diseases and COPD was carried out in the updated DIANA Tools, miRTarBase-bio, and TargetScan databases (available online links: https://bio.tools/mirtarbase and http://www.targetscan.org) obtaining the correlation (R), (miRNA-Target expression profile). The serum concentration of the human protein histone deacetylase 4 (HDAC4), was assessed using the ELISA test kit (Human Histone Deacetylase 4/HDAC4; EKU04816; Biomatik, Wilmington, DE, USA), following the manufacturer’s instructions.

### 2.8. Statistical Analysis

The demographic and clinical characteristics of the study populations were expressed as means and standard deviations. Statistical analysis was performed using Student’s *t*-test. RT-qPCR was analyzed by relative quantification (ΔCt method), differential expression of miRNA and quantification of the HDAC4 protein were also evaluated by Student’s *t*-test. Additionally, the correlation between protein levels and DLCOsb %P was calculated. *p*-values <0.05 were considered statistically significant. Analyses were performed using the GraphPad statistical package version 6.01 (GraphPad Software, Inc., La Jolla, CA, USA). 

## 3. Results 

### 3.1. Characteristics of Women in the Study

The clinical data of the groups are shown in Table 1. Women with COPD by biomass were older, shorter, and had a higher DLCOsb %P compared with those with COPD by tobacco (*p* < 0.01). The weight, BMI, FEV1%P, FEV_1_/FVC ratio, and 6MWT; showed no significant differences between groups. 

### 3.2. miRNA-22 was Upregulated in COPD by Biomass Related to COPD by Smoking

The expression of miR-22 in serum measured in PCR arrays between COPD-BS and COPD-TS had a fold change of −528.96 (*p* value 0.004), showing that miR-22 was negatively regulated in women with COPD-BS compared to COPD-TS. This was later confirmed by the technique of RT-qPCR, and statistical analysis was performed using Student’s *t*-test (*p* < 0.05; Figure 1).

### 3.3. Serum HDAC4 is Higher in COPD by Biomass that COPD by Smoking

The serum concentration of HDAC4 was quantified by ELISA and compared between COPD-TS and COPD-BS (*n* = 25), showing an increase in COPD-BS compared to women with COPD-TS (*p* < 0.01, Figure 2). 

### 3.4. HDAC4 Positively Correlated with DLCOsb %P in COPD 

With the aim of analyzing whether there is an association between the serum concentration of HDAC4 with DLCOsb %P, a correlation coefficient between both variables was evaluated, displaying a positive correlation between the serum level of HDAC4 and DLCO%P (* *p* < 0.01, *r* = 0.94; Figure 3). 

## 4. Discussion

Our objective was to analyze the miR-22-HDAC4-DLCO axis among patients with COPD by tobacco and biomass smoke. We found that miR-22 was down-regulated in patients with COPD-BS compared to patients with COPD-TS. On the other hand, the HDAC4 protein was increased in patients with COPD-BS compared to patients with COPD-TS, and we observed a positive correlation between protein levels and the levels of DLCOsb. 

Our analysis was restricted to women, because women in Mexico tend to be more exposed to BS than men because they are more likely to prepare meals, as a domestic role. Therefore, is less likely than men develop COPD from this cause. COPD-diagnosed women have an inflammation profile that is higher than normal control levels, as is evidenced by higher plasma CRP (C-reactive protein), glucose, and several inflammatory mediators, such as adipokines, incretins, and peptide hormones [18]. This supports the fact that COPD-BS and COPD-TS could represent different phenotypes of COPD [2,3]. It is important to note that in this study, most of our patients were hypoventilators and hypoxemic, all women showed a FEV1% prediction due to the height of Mexico City (~2240 m above sea level), which has effect on the functional capacities of the pulmonary and cardiovascular systems, and consequently the oxygen flow rate [19,20]. 

The miR-22-HDAC4-Th17 axis has been previously studied by Corry et al., who showed that exposure to smoking caused a down-regulation of miRNA 22, with a decrease in HDAC4 and an increase in TH17 immune response, which caused an increase in IL-6 and TNF and resulted in the disruption of the alveolar walls, and the formation of emphysema [13]. Our findings for COPD-TS are consistent with those related to miR-22 -HDAC4-Th17 proposed by Corry et al. To our knowledge, this is the first study that depicts the participation of the miR-22 axis in emphysema formation in patients with COPD-TS. 

HDAC4 is a key member of class IIa HDACs and it is expressed in multiple tissues [21]. HDACs have been previously associated with the pathogenesis of emphysema. Suppressed HDAC activity in lung biopsies and lung macrophages of smokers with emphysema correlates positively with the severity of the disease [22]. A further study of HDAC4 inhibitors in vivo and in vitro suggested that HDACs have an anti-inflammatory role in emphysema [23]. We found that HDAC4 in patients with COPD-TS was lower than in patients with COPD-BS, which is consistent with in vitro and in vivo studies. This observation sheds light into the reasons why COPD-TS produces emphysema, as opposed to COPD-BS. In this axis, TH17 is a critical point that we were unable to measure. However, it has been reported that patients with COPD-BS had a lower TH17 response associated with a Th2-type lymphocyte response, compared to COPD-TS patients, whose pathogenic mechanism is dominated by Th1-type associated with higher Th17 cytokine production [24,25]. From this observation we can infer that high levels of HDAC4 mediated by miR-22 down-regulation could be involved in the reduced TH17 response found in patients with COPD-BS, leading to a lower inflammatory response and a decrease in emphysema, in comparison with patients with COPD-TS. 

Some limitations of our study must be mentioned. Our patients did not undergo chest CT scans to evaluate emphysema, which are able to quantitate lung zones with low density suggestive of emphysema. Instead, we took DLCOsb as a proxy for emphysema, due to a good correlation of DLCOsb with the presence of emphysema [26,27]. Although, this takes in consideration a recent study, which by means of a multivariable analysis, it was shown that every 10% predicted decrease in DLCOsb was associated with symptoms and quality of life, physical function, and exercise performance measure by 6MWT, severe exacerbation rate, and worse morbidity [28].

Additional differences may influence phenotypic characteristics of COPD-BS and COPD-TS, such as socio-economic level and level of education, which are both lower in COPD-BS. Having Amerindian genes is more commonly associated with COPD-BS and that may be linked with genetic susceptibility, contrasting with increased exposure to particulate matter in tobacco smokers and also with more exposure to air pollution in the city, which could influence our results. Moreover, usually there is more intense exposure to PM2.5 in tobacco smokers compared to individuals exposed to biomass smoke COPD [29]. The composition of tobacco biomass and wood or fuel biomass may explain differences in the presentation of both forms of COPD [2,3,4,6,25]. 

## 5. Conclusions

We depicted the potential involvement of the miR-22-HDAC4-DLCO axis on the pathogenesis of COPD associated with tobacco smoke, contrasting with that due to biomass smoke inhalation, which may lead mechanistically to two phenotypes of COPD well-known to clinics from developing countries, COPD-BS and COPD-TS. 

## 6. Patents

None to declare.

## Figures and Tables

**Figure 1 biomolecules-09-00837-f001:**
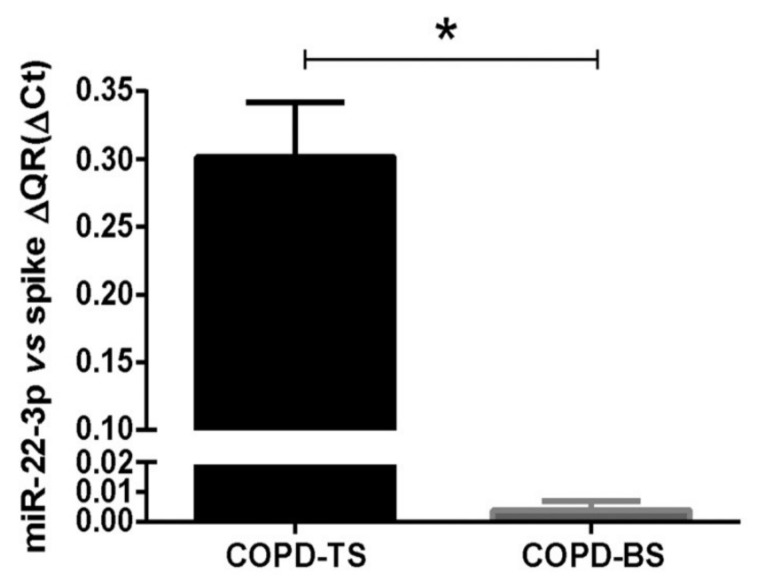
miRNA-22 is upregulated in chronic obstructive pulmonary disease (COPD) by biomass related to COPD by tobacco. The cDNAs were obtained using the RT kit and TaqMan Universal Master Mix II with UNG (Uracil-N-glycosylate; Applied Biosystems-Thermo Fisher Scientific). The miRNA validated by RT-qPCR was miR-22-3p, downregulated in COPD-BS compared to COPD-TS (*n* = 25), (*p* < 0.05, Figure 1). The data are presented as ΔCt (2^-CT gene of interest − CT internal control^) values. Student’s independent sample *t*-test was used. The analyzes were performed using the GraphPad statistical package version 6.01 (GraphPad Software, Inc., La Jolla, CA, USA). * *p* < 0.01. Abbreviations: COPD-BS, COPD biomass smoke exposure; COPD-TS, COPD by tobacco smoke exposure.

**Figure 2 biomolecules-09-00837-f002:**
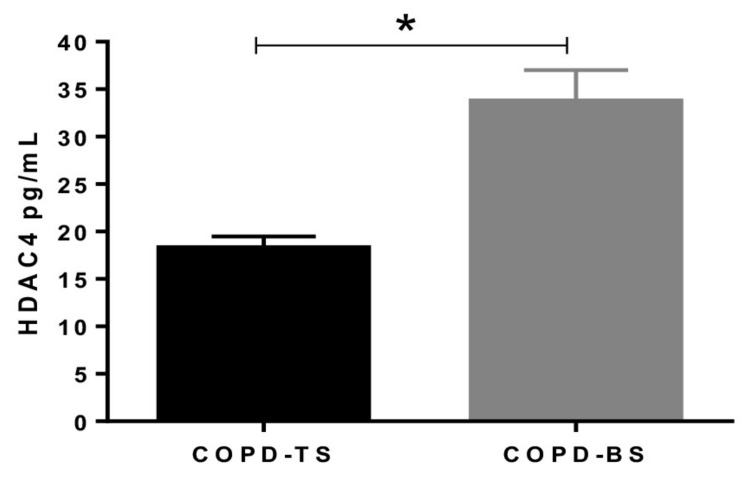
Serum HDAC4 protein is higher in COPD by biomass than COPD by smoking. The protein HDAC4 level was quantified in serum by ELISA and was expressed in pg /mL (*n* = 25). * *p* < 0.01. Abbreviations: COPD-BS, COPD biomass smoke exposure; COPD-TS, COPD by tobacco smoke exposure, HDAC4; Histone Deacetylase 4.

**Figure 3 biomolecules-09-00837-f003:**
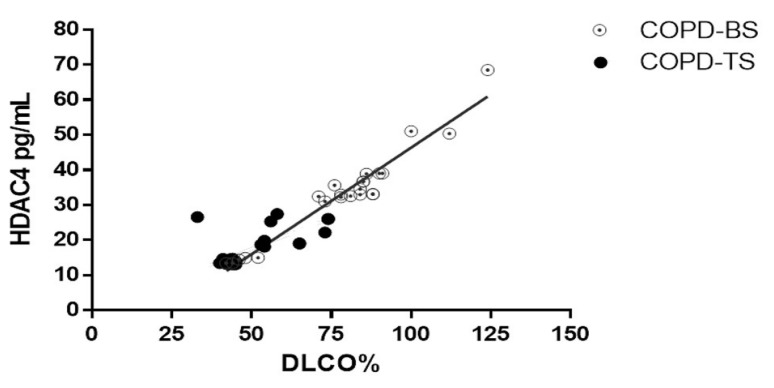
HDAC4 positively correlated with DLCOsb in COPD. Relationship between DLCOsb %P and HDAC4 protein in the COPD-TS and COPD-BS groups (*n* = 25). *r* = 0.94. * *p* < 0.01. Abbreviations: COPD-BS, COPD biomass smoke exposure; COPD-TS, COPD by tobacco smoke exposure; HDAC4, Histone Deacetylase 4; DLCOsb %P, Single-breath diffusing capacity of the lung for carbon monoxide (% predicted).

**Table 1 biomolecules-09-00837-t001:** Clinical and demographic characteristics.

**Variable**	**COPD-TS**	**COPD-BS**
**Characteristics**		
**Age (years)**	66.57 ± 6.37	73.27 ± 8.69*
**Height (cm)**	155.31 ± 7.57	146.52 ± 5.69*
**Weight (Kg)**	59.52 ± 12.13	56.25 ± 10.82
**BMI (Kg/m2)**	24.79 ± 5.78	26.33 ± 5.04
**Exposure Characteristics**		
**Pack-years of smoking**	36.62 ± 23.1	0
**Hour-years of biomass smoke exposure**	0	366.88 ± 219.3
**Physiological characteristics**		
**FEV_1_%P**	39.85 ± 5.36	39.92 ± 6.57
**FEV_1_/FVC ratio**	45.63 ± 11.41	45.23 ± 10.08
**DLCOsb %P**	50.96 ± 11.88	78.45 ± 20.47*
**6MWT (m)**	286.47 ± 174.23	318.91 ± 102.99
**GOLD grades**	Case numbers (%)	
**III**	16 (64)	20 (80)
**IV**	9 (36)	5 (20)

Data are expressed as means ± SD. Abbreviations: BMI, body mass index; COPD-BS, chronic obstructive pulmonary disease (COPD) exposure to biomass smoke, COPD-TS, COPD exposure to tobacco smoke. FEV1%P, forced expiratory volume in the 1st second (% predicted); FVC, forced vital capacity; FEV1/FVC ratio, forced expiratory volume in the 1st second (% predicted)/ forced vital capacity ratio; DLCOsb %P, Single-breath diffusing capacity of the lung for carbon monoxide (% predicted); and 6MWT (m), 6-min walk test (meters). The statistical analysis was carried out by Student’s *t-*test. * *p* < 0.01.

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
