# Peer review of "Participation of the miR-22-HDAC4-DLCO Axis in Patients with COPD by Tobacco and Biomass"

_biomolecules, 2019, doi:10.3390/biom9120837_

Round 1

Reviewer 1 Report

The topic of this manuscript is very interesting. The experimental protocol has been rationally designed and carefully carried out. Methods are appropriate, and results are clearly described and properly discussed. The quality of written English should be improved.

Author Response

RESPONSE TO REVIEWER ONE

The topic of this manuscript is very interesting. The experimental protocol has been rationally designed and carefully carried out. Methods are appropriate, and results are clearly described and properly discussed. The quality of written English should be improved.

Response 1: We appreciate the review and the comments.

We all checked spelling and grammar of our manuscript and share it with colleagues fluent in English. We acknowledge though, that it is difficult for us to identify language weaknesses in our manuscript. All changes are label in rex text.

Reviewer 2 Report

GENERAL COMMENTS

The current study tries to demonstrate that the miR-22-HDAC4-DLCO axis is differentially modified in COPD patients induced by cigarette smoking from those exposed to biomass. The study is of interest and in general it has been well written. However, I have several comments that are summarized below.

SPECIFIC MAJOR COMMENTS

1) The study design should be better defined. Was this a prospective controlled study? On what basis were the patients recruited? Which were the established inclusion criteria? Were the patients subdivided into the two groups ad hoc or post hoc? On what basis? This is crucial information in order to assess the quality of the investigation and also for the sake of reliability of the study findings.

2) CT scan measurements of the degree of emphysema should be reported for each of the groups. In which group of patients was emphysema more severe?

3) In Table 1, exercise tolerance should be provided for each group of patients. This is relevant clinical information that allows us to better understand the severity of the emphysema in each group of patients.

4) Discussion is excessively short and merely descriptive of the study findings. It should be extended in order to further interpret the study findings in contrast to previous studies of similar characteristics. Moreover, the potential downstream mechanisms that are regulated by the miR-22-HDAC4 axis should be explained and discussed.

5) Which are the clinical and functional implications of the differential regulation of this axis in the patients?

6) How can alterations in the expression of this axis lead to a greater deterioration of the lungs in the patients, especially in those exposed to tobacco? Which are the mechanisms?

7) The conclusions are far overreaching and should be written on the basis of the actual findings, especially when the mechanisms linking emphysema extension with the levels of the miR22-HDAC4 axis have not been explored in any of the patients.

Author Response

RESPONSE TO REVIEWER TWO

GENERAL COMMENTS

The current study tries to demonstrate that the miR-22-HDAC4- DLCO axis is differentially modified in COPD patients induced by cigarette smoking from those exposed to biomass.

The study is of interest and in general it has been well written. However, I have several comments that are summarized below.

Response: We appreciate your review and your comments. Most of them were followed.

SPECIFIC MAJOR COMMENTS

1) The study design should be better defined. Was this a prospective controlled study? On what basis were the patients recruited? Which were the established inclusion criteria? Were the patients subdivided into the two groups ad hoc or post hoc? On what basis? This is crucial information in order to assess the quality of the investigation and also for the sake of reliability of the study findings.

Response 1: These considerations are important. The groups of COPD separated in exposed to biomass smoke and tobacco smokers were defined since the cohorts were started and this comparison has been the theme of several manuscripts from our group for a long time. We consider this comparison very relevant for the mechanisms of COPD generation, usually considered a disease linked predominantly with smoking. It is then a prospective study, a cohort, and we included consecutive patients accepting to participate and had measurements of markers included in our study.

These methodological details are now considered in 2. Material and methods, 2.1. Study population section.

Page 2; lines 78-92.

2) CT scan measurements of the degree of emphysema should be reported for each of the groups. In which group of patients was emphysema more severe?

Response 2: Thanks for this observation. The analysis of CT, especially a quantitative assessment of CT would be a relevant addition to our study. Unfortunately, a minority of the patients had CT and quantitative CT, information difficult to analyze and to compare between groups. That is the reason we are utilizing a functional indicator of emphysema, the DLCO. The routine use of CT in patients with COPD is not considered essential, the criteria followed by clinicians in our institute, in comparison with spirometry now essential for diagnosis. With recent studies such as the COPD gene, CT scanning would become probably necessary even from the clinical point of view. 

3) In Table 1, exercise tolerance should be provided for each group of patients. This is relevant clinical information that allows us to better understand the severity of the emphysema in each group of patients.

Response 3: Information on exercise, 6-minute walking test, is now in the manuscript. It is important to recognize that all are residents at moderate-high altitude, ~2240 m above sea level, and this reduces the physical performance, and oxygenation, forming an interesting group, as there is a substantial population residing at moderate-high altitude.

See Table 1 in page 5

4) Discussion is excessively short and merely descriptive of the study findings. It should be extended in order to further interpret the study findings in contrast to previous studies of similar characteristics. Moreover, the potential downstream mechanisms that are regulated by the miR-22-HDAC4 axis should be explained and discussed.

Response 4: Discussion was extended considering the role of miR-22HDAC4 axis in the pathology of COPD secondary to biomass smoke, comparatively with those secondary to Tobacco smoking.

5) Which are the clinical and functional implications of the differential regulation of this axis in the patients?

6) How can alterations in the expression of this axis lead to a greater deterioration of the lungs in the patients, especially in those exposed to tobacco? Which are the mechanisms?

Response 5 and 6: The discussion was extended and improved, especially in the proposed mechanisms for differences between COPD exposed to smoking and biomass smoke. It is clear that smokers develop a more severe airflow obstruction and emphysema than patient with COPD exposed to biomass smoke. In fact, both tobacco and biomass fuel are forms of biomass, and smokes have a great similarity. The differences of components of smoke could be one factor, and one clear difference is nicotine in tobacco, but also the magnitude of exposure to PM2.5 is considerably higher in tobacco smokers than in women exposed to biomass smoke. Simply, to develop clinical emphysema a great exposure to smoke and PM2.5 is required and frequently achieved by smokers but not by those exposed to biomass. In post-mortem studies, women may have different degrees of emphysema on microscopy, always less than in smokers, but not when tested with CT, and usually do not develop a low DLCO.  The proposed mechanisms in our manuscript may explain these clinical differences at least partially.

Page 6; line 211

Page 7; lines 212-221, 237-243, and 246-256

7) The conclusions are far overreaching and should be written on the basis of the actual findings, especially when the mechanisms linking emphysema extension with the levels of the miR22-HDAC4 axis have not been explored in any of the patients.

Response 5: The conclusions were modified and moderated. Thanks for the comment.

New conclusions are:  We depicted the potential involvement of miR-22-HDAC4-DLCO axis on the pathogenesis of COPD associated with tobacco smoke, contrasting with that due to biomass smoke inhalation, that may lead mechanistically to two phenotypes of COPD well known to clinics from developing countries: COPD-BS and COPD-TS.

Page 7; lines 258-261

Round 2

Reviewer 2 Report

The authors have addressed my comments. I have no additional comments.